

# Location and distribution of micro-inclusions in the EDML and NEEM ice cores using optical microscopy and in-situ Raman spectroscopy

Jan Eichler[1,2], Ina Kleitz[1], Maddalena Bayer[1], Daniela Jansen[1], Sepp Kipfstuhl[1], Wataru Shigeyama[3,4], Christian Weikusat[1], and Ilka Weikusat[1,2]

[1]Alfred Wegener Institute Helmholtz Centre for Polar and Marine Research, 27568 Bremerhaven, Germany
[2]Department of Geosciences, Eberhard Karls University Tübingen, 72074 Tübingen, Germany
[3]Department of Polar Science, SOKENDAI (The Graduate University for Advanced Studies), 10-3 Midori-cho, Tachikawa, Tokyo, 190-8518, Japan
[4]National Institute of Polar Research, 10-3 Midori-cho, Tachikawa, Tokyo, 190-8518, Japan

*Correspondence to:* J. Eichler (jan.eichler@awi.de)

**Abstract.** Impurities control a variety of physical properties of polar ice. Their impact can be observed at all scales - from the microstructure (e.g., grain size and orientation) to the ice sheet flow behavior (e.g., borehole tilting and closure). Most impurities are likely to form micrometer-sized second phase inclusions. It has been suggested that these particles control the grain size of polycrystalline ice by the pinning of grain boundaries (Zener pinning), which should be reflected in the distribution

of the inclusions in relation to the grain boundary network. We used an optical microscope to generate high-resolution large-scale maps ($3\,\mathrm{\mu m\,pix^{-1}}$, $8 \times 2\,\mathrm{cm^2}$) of the distribution of micro-inclusions in four samples from the EDML (Antarctica) and NEEM (Greenland) polar ice cores. The in-situ positions of more than 5000 μ-inclusions have been determined. A Raman microscope was used to confirm the extrinsic nature of a sample proportion of the mapped inclusions. A superposition of the 2D grain boundary network and μ-inclusion distributions show no significant correlations between grain boundaries and

μ-inclusions. In particular, no signs could be found of grain boundaries harvesting μ-inclusions, no evidence of μ-inclusions inhibiting grain boundary migration by slow mode pinning could be detected. Consequences for our understanding of the impurity effect on ice microstructure and rheology are discussed.

## 1 Introduction

Polar meteoric ice is one of the purest materials on Earth. Impurity mass concentrations in the Antarctic ice sheet vary between
15 ppbm during warm periods and ppmm in the most dusty layers in cold periods (Faria et al., 2010). Values for the Greenland ice sheet are approximately ten times higher. Impurities enter the ice sheet during precipitation in form of terrestrial dust, salt particles and other aerosols e.g. from bio-activity, ocean, volcanoes or combustion, or in form of gas inclusions from air bubbles. Trace substances are a subject of intense ice core studies for many reasons. Their chemical and isotopic compositions can be linked to atmospheric and climatic processes (climate proxies), their analysis provides an important insight into processes and
20 constraints of the past of Earth's climate (Dansgaard et al., 1982; GRIP Project Members, 1993; EPICA community members,



2004, 2006; Kuramoto et al., 2011; Wegner et al., 2015). When linked to geologic events (Sigl et al., 2015), they can serve as absolute chronological markers for ice core dating (sulphate and tephra layers from volcanic eruptions, Gow and Meese, 2007). And finally, many material properties of ice are controlled or modulated by its impurity content. This is the case for the dielectric constant and electrical conductivity (Wolff et al., 1997; Stillman et al., 2013; Taylor et al., 1993; Wilhelms et al., 1998), but also mechanical properties of ice - such as creep behavior and viscosity - which are of great interest with respect to ice rheology and ice sheet dynamics. The influence of various impurities on deformation rate has been observed in laboratory tests (e.g., Dahl-Jensen et al., 1997) as well as in the field from borehole tilting and closure (e.g., Fisher and Koerner, 1986). The link between impurity content and strain rate becomes most evident when comparing glacial ice with ice from warm periods. Observations show that the impurity-rich ice-age ice deforms on average 2.5 times faster in simple shear (Paterson, 1991). Several explanations for this effect have been proposed, however the responsible mechanisms still remain under discussion. Since impurity-rich ice usually develops a different microstructure - e.g., stronger crystal preferred orientation (CPO), smaller grains (Paterson, 1991; Cuffey et al., 2000) - the higher strain rates may result from an interplay between impurities, microstructure and recrystallization.

With increasing interest in the impurity content, analysis methods have been refined over the past years. Continuous flow analysis (CFA) became a standard part of ice core processing (Kaufmann et al., 2008) owing to its effectivity and high depth resolution. However, the aim to understand impurity-related processes in polycrystalline ice requires the application of approaches dedicated to the solid state. Thus, advanced analytical techniques, such as scanning electron microscopy (SEM), Raman spectroscopy or laser ablation inductively coupled plasma mass spectrometry (LA-ICPMS) became popular by glaciologists. However, each of these methods brings its specifications and constraints and the results often lead to contradictory conclusions. Comparative studies may be necessary to clear these discrepancies in future.

The composition, form, location and distribution of impurities are different in ice compared to liquid water. Due to the electric dipole moment of the $H_2O$ molecule, water is a good solvent. On the contrary, the ice crystal is not, and will reject most extrinsic substances out of the matrix forming a second phase (Petrenko and Whitworth, 1999). Only few elements are theoretically able to be incorporated into the ice lattice. According to Jones (1967) and Jones and Glen (1969) $F^-$, $Cl^-$, $K^+$ and $NH_4^+$ in low concentrations are able to substitute oxygen atoms or enter the ice lattice interstitials. In both cases they would introduce ionic and Bjerrum-defects and significantly affect the charge carrier density of the ice crystal. Since protonic defects can alter dislocation mobilities, ice doped with these species manifests higher strain rates than pure ice (Glen, 1968). However, the doping experiments by Jones and Glen probably represent upper bounds with respect to natural ice which is formed from snow flakes inheriting impurities from atmospheric processes. Another widely discussed form of ice impurities was proposed by Wolff and Paren (1984) who interpreted DC-conductivity as caused by conduction through acidic environments along grain boundaries and triple junctions. The authors suggested that acids like HCl, $HNO_3$ and $H_2SO_4$ would concentrate at grain boundaries lowering the eutectic point of the solute and thus forming liquid veins. The existence of an acidic vein network was supported by Mulvaney et al. (1988) who found sulfur at three triple junctions using energy-dispersive X-ray spectroscopy (EDX), and Fukazawa et al. (1998) who measured a sulfate peak in a triple junction using a Raman microscope. However, other Raman-spectroscopic studies by Ohno et al. (2005, 2006) and Sakurai et al. (2011) found most (or all) sulfates in form





of salt particles. In contrast, EDX experiments by Cullen and Baker (2001), Barnes et al. (2002), Barnes (2003), Baker et al. (2003) and Iliescu and Baker (2008) found traces of sodium, chlorine and sulfur in filaments, which would grow out of grain boundaries after controlled surface sublimation of natural ice samples. There is no consensus yet about the abundance and relevance of impurity segregation to grain boundaries.

In terms of mass fraction, most impurities form second phase inclusions, typically of the size of few micrometers - in the following called μ-inclusions. Water-insoluble dust particles as well as water-soluble particulate salts are the most abundant μ-inclusions. Their concentrations are highly variable along the ice cores. For instance, dust concentrations in the EDML ice core measured by continuous flow analysis (CFA, Wegner et al., 2015) vary between $10^3$ and $10^5$ particles per milliliter. Strata with high concentrations of μ-inclusions are visible in ice cores and are often called "cloudy bands" (Svensson et al.,

2005). μ-inclusions appear under an optical microscope as dark spots near the optical detection limit - commonly referred to as "black dots" (see Fig.1). When using optical microscopy it is impossible to examine their state of aggregate, shape, color or composition, and thus the term "black dot" not only reflects their visual appearance but also our uncertainty about their nature and composition.

High concentrations of μ-inclusions are usually associated with certain changes in ice microstructure. Cloudy ice exhibits

smaller grain sizes and often stronger CPO than clean ice. Since grain growth involves grain boundary migration, impeding grain boundary movement by μ-inclusions would consequently have a grain-size reducing effect. The attractive force between second phase inclusions and grain boundaries is known from material sciences and has been first modeled by Zener (in Smith, 1948) - Zener pinning. When a migrating grain boundary passes a particle, its energy is reduced by the portion of the cross sectional area. Depending on the grain boundary driving force, the pinning pressure and the mobility of the particles, grain

boundary can be completely stopped (pinned), free itself after a while from the particles and continue its motion, or drag the particles with it. Alley et al. (1986a, b) reviewed the Zener theory and available data with respect to the pinning effect on normal grain growth in cold ice. They differentiate between a low velocity regime, where the grain boundary is pinned by the impurities, and a high velocity regime, where the grain boundary continues its motion leaving the impurities behind. Alley et al. (1986b) conclude that pinning on μ-inclusions occurs in high velocity regime and the concentration of microparticles is

too low in general to significantly affect grain growth (the only exceptions are tephra layers from volcanic eruptions). Durand et al. (2006) simulated grain size evolution along the Dome Concordia ice core by modeling normal grain growth controlled by pinning on dust particles. They suggest that dust particles will indeed impede grain growth if they concentrate at grain boundaries. However, a direct proof for such a particle distribution was not provided, since grain boundaries could not be imaged. On the contrary, Raman-measurements of μ-inclusions in the Dome Fuji ice core, presented by Ohno et al. (2005,

2006), showed that major part of the particles (mostly sulfate salts) were located in grain interiors. Faria et al. (2010) evaluated relevant microstructure and impurity data concerning the integrity of the EDML ice core. According to their report, black dots in the microstructure mapping images do not accumulate along grain boundaries for depths down to 2300 m. Only below this depth, and particularly in the deepest 200 m of the core, accumulations at grain boundaries and on the surface of clathrate hydrates were observed.



So far, studies on spatial distributions of µ-inclusions in ice were based on discrete observations of few dust or salt particles. In order to discuss general concepts, such as the formation of the CFA signal, predominant form of impurities in-situ and their effect on recrystallization (e.g. grain boundary pinning), more systematic and statistically relevant approaches are necessary. The aim of this study is to provide a more detailed insight into the in-situ concentrations and distributions of µ-inclusions. We

mapped over 5000 µ-inclusions within four different ice samples from polar ice cores. Overall and local concentrations are estimated and compared with the available CFA data. Special attention is payed to the correlation between µ-inclusions and the grain boundary network. We observe no accumulations of µ-inclusions along grain boundaries and argue that only fast mode pinning is active in the samples.

## 2 Methods and sample material

The state in which impurities are included in ice is different from their form in melt water (CFA). In order to reveal their distribution and in-situ form, more data based on direct measurements of ice samples are needed. Optical microscopy and Raman spectroscopy provide the opportunity to explore the interior of ice samples in a non-destructive way. This is a significant advantage over surface-based methods (e.g., SEM), where contamination and sublimation-related redistribution of impurities have to be considered (e.g., Barnes, 2003). We use microstructure mapping to create large area maps ($8 \times 2$ cm$^2$) of the

specimens surfaces and interiors. This microscopical technique was introduced by Kipfstuhl et al. (2006) and it enables us to pinpoint visible µ-inclusions as well as grain and subgrain boundaries. We use a Raman microscope to investigate the composition of selected µ-inclusions and to ensure that we are dealing with real second phase inclusions.

As the lower impurity content in warm period ice provide a better chance to observe and characterize the majority of µ-inclusions over a large sample area and analyze their correlation with the microstructure, we focused our study on samples

from warmer periods. Furthermore, one of the very few studies which found evidence for a connection of spatial distribution of impurities along grain boundaries reported this evidence as a characteristic of "clean" interstadial ice (Della Lunga et al., 2014). Four samples were analyzed - two from the EDML ice core (Antarctica) and two from the NEEM ice core (Greenland). The EDML samples (EDML-2371.4, EDML-2371.9) originate from 2370 m depth which corresponds to the early Eemian (MIS5.5). The NEEM samples (NEEM-1346.2, NEEM-1346.5) originate from the Holocene, 740 m below surface and around

4000 years b2k (Rasmussen et al., 2013).

### 2.1 Impurity maps

The sample preparation has been described by Kipfstuhl et al. (2006). Specimens of the thickness of around 1 cm are cut with a band saw (see cutting plan in Fig.3d). Both surfaces are polished with a microtome knife and exposed to air for a few hours. Sublimation smoothens the surface and creates grooves at sites of high energy, where grain and subgrain boundaries intersect

the surface. In this way 2D-maps of grain boundary networks and subgrain structures can be created (Binder et al., 2013). When focusing into the ice volume and choosing transmission light mode, surface features such as etching grooves fade out but other objects inside the sample come into focus. Typical features are gas inclusions (air bubbles and clathrate hydrates), relaxation





features (secondary bubbles, plate-like inclusions) and μ-inclusions (see examples in Fig.1b,c and Fig.2c,d). μ-inclusions appear as dark spots near the resolution limit of the microscope.

We use an optical microscope (Leica DMLM) with a CCD camera (Hamamatsu C5405), frame grabber and a software-controlled x-y stage. Ice samples of up to 10 cm side length can be scanned at the resolution of 3 μm pix$^{-1}$. The final microstructure map is stitched from up to 1500 individually captured photomicrographs. We combine two of these scans for each sample - a microstructure map focused onto the surface to reveal grain boundaries, and an impurity map focused ca. 300 μm into the sample volume to visualize μ-inclusions (see Fig.1a). Such a stack of maps allows us to study inclusions in direct relation to the grain boundary network. Since μ-inclusions are mapped within the sample volume, contamination is not an issue. Due to the obliquity and 3D shape of grain boundaries, their positions inside the sample are not exactly the same as the etching grooves on the surface. To keep this uncertainty low the second scan must be focused close below the surface. Black dots in impurity maps were detected manually. The low contrast and size of black dots in the images did not support the application of automatic detection filters. Visual detection and manual counting of μ-inclusions is a time-consuming process which requires a certain amount of patience and discipline. Since it is based on observer's subjective judgement, the results may contain an observer-dependent variance. The impurity maps presented in this study were generated by three observers independently, however, using the same criteria. Partial comparison of the results at overlapping regions showed a good consistency within the data and thus confirmed the observer-dependent factor being minimal.

## 2.2 Raman spectroscopy

A confocal Raman microscope was used to analyze the composition and mineralogy of selected μ-inclusions. The AWI cryo-Raman system consists of a WITec alpha 300 M+ with an UHTS 300 spectrometer and a Nd:YAG laser (532 nm) set up in the cryolab at $-15\,°C$ (Weikusat et al., 2015). Within the scope of this paper, the measured Raman spectra shall serve as a proof, that the mapped black dots are in fact real second phase μ-inclusions. A detailed analysis of the detected minerals is being prepared for publication.

## 3 Results

Microstructure and impurity maps were generated for the four samples: EDML-2371.4, EDML-2371.9, NEEM-1346.2, NEEM-1346.5. We localized 5784 μ-inclusions in total (in Fig.2,3,4,5 marked with yellow circles). Their size of 2–3 pixels in diameter would correspond to 6–9 μm, however their appearance as "black dots" is probably produced by optical effect of much smaller particles of the typical dust size ca. 1–2 μm (Wegner et al., 2015). Raman measurements and correlation with CFA dust and $Ca^{2+}$ peaks however confirm our assumption that these features are real μ-inclusions.

## 3.1 Raman spectroscopy

In order to ensure that the mapped black dots are real second phase μ-inclusions, small fractions of them (between 20 and 40 inclusions) in each sample were selected for the Raman analysis. Using a 50x lens and confocal mode, a great majority (around



90%) of the μ-inclusions could be found again in the Raman microscope. Additionally, their z-positions could be measured due to the confocal setting. Z-positions of μ-inclusions are Gaussian-distributed around the focal plane with a half maximum width of 200 μm. We accept this value as the depth of field of the mapping microscope and use it for the calculation of the volume fraction of the impurity maps.

Around 70% of analyzed μ-inclusions showed a Raman spectrum of sufficiently high intensity to separate it from the overall present ice spectrum. The quality of obtained Raman signal depends on several factors. The most limiting ones are the size of the μ-inclusion, the path length of the laser beam through the crystal, the quality of polished surface and the acquisition time. Due to time constrainst given the large amount of measured points, integration time was in the range 5–10 seconds with 10 repetitions. Most of the measured inclusions in EDML-2371.4 and EDML-2371.9 - around 40 spectra - could be identified as

sulfate salts (see example spectrum in Fig.2e). The NEEM samples showed a higher content of water-insoluble inclusions, such as quartz and black carbon. A detailed description of the composition statistics is still in progress and will be shown elsewhere. However, the Raman measurements show that the counted black dots are chemical impurities and thus verify microstructure mapping as a valid method to map the visible in-situ impurity content.

### 3.2   Concentration

The numbers of counted μ-inclusion per sample are shown in Tab.1. Knowing the depth of field being 200 μm we calculated the volume fraction of the mapped area. In this way, the average concentration of μ-inclusions per ml of water equivalent could be estimated. Comparison with the CFA dust concentration shows a clear difference between the deep EDML ice and the shallow NEEM ice. In EDML, the number of visible μ-inclusions in-situ is 2–3 times higher than the content of insoluble particles (dust) in the melt water. In contrast the amount of μ-inclusion in the NEEM samples is comparable or even less than the CFA

dust concentration.

### 3.3   Distribution

#### 3.3.1   General spatial distribution

The distributions of μ-inclusions within the four samples are highly inhomogeneous. Similarly to ice-age ice (cloudy ice) also the cleaner Holocene and Eemian ice (MIS5.5) contains horizontal layers of increased impurity concentrations (analogous to

cloudy bands) instead of μ-inclusions being distributed homogeneously.

    In the EDML-2371.9 sample more than 50% of the counted black dots can be allocated to one of these horizons. We can distinguish a sharp double horizon at 2370.95 m (Fig.3a, central part of the sample). This sharp double layer correlates with a cloudy band in the visual stratigraphy scanning image (Fig.3b) and with dust and calcium peaks in the CFA profile. Around 9 mm below, there is another (disrupted) impurity layer at 2370.96 m and one more at the bottom part of the section around

2370.99 m. In the sample EDML-2371.4 (Fig.4) the layers are not as sharp as in 2371.9 but horizons of higher and lower concentrations are clearly visible. The NEEM samples (Fig.5) also show horizontal layering, however not so well defined and more continuous.





While the distribution of μ-inclusions on the cm scale occurs in horizontal or sub-horizontal layers, on the μm scale they are often aggregated in clusters. These groups of two or more adjacent black dots are typical for both - regions with low concentration as well as high-impurity layers.

### 3.3.2 Distribution with respect to microstructure

Grain boundary grooves from microstructure mapping of the sample surfaces are highlighted in Fig.3,4,5 with blue lines of a thickness of 300 μm. This width was chosen to represent the possible GB-position-error rising from the unknown inclination of the grain boundary below the sample surface. In Tab.2, μ-inclusions located within this blue range are considered as potentially interacting with the grain boundary. This is a generous assumption considering the fact that a real grain boundary is not thicker than only a few nm. Tab.2 shows the fraction of μ-inclusions found within a range of 300 μm around a grain boundary. The percentages clearly show that the vast majority of μ-inclusions is located away from grain boundaries. In EDML-2371.4, 89% of μ-inclusions are situated further than 150 μm away from any grain boundary. In EDML-2371.9, the percentage of 93% of μ-inclusions located not in the vicinity of grain boundaries is even higher. Slightly higher percentages of μ-inclusions related to grain boundaries are observed in the NEEM samples: 24% and 15%. The "grain-boundary region" in both sample types has been kept constant for observational reasons such as same imaging resolution, similar focus depth of impurity maps and unknown grain boundary inclination. However the NEEM samples show a significantly smaller grain size (mean radius 1.5 mm) compared to the EDML samples (mean radius 2.5 mm, Weikusat et al., 2009).

In general no correlation between μ-inclusions and grain boundaries could be detected in any of the analyzed samples. Instead, the distinctive horizons 2 and 3 in EDML-2371.9 are located inside big grains, several millimeters away from the nearest grain boundaries.

In both EDML samples we observe high accumulations of secondary gas inclusions along grain boundaries, which are formed due to relaxation of the material (Weikusat et al., 2012). Their high densities allow us to partly reconstruct the 3-dimensional shape of the grain boundary just by means of the micro-bubbles (Fig.2,4).

The concentration of μ-inclusions and clusters seems not to depend on shape, size or crystal orientation of individual grains (Fig.3c). Black dots follow sub-horizontal layering as mentioned above, rather than any microstructural feature.

## 4 Discussion

### 4.1 Impurities in form of μ-inclusions

We used a microscopic method to map visible impurities within the ice sample volume. Since μ-inclusions are mapped in-situ, surface contamination of the sample does not affect the results. The method is limited mainly by the small sizes of inclusions, which are close to the resolution limit of the system. Objects smaller than this limit are virtually not resolved and thus a fraction of small-sized μ-inclusions may be excluded from the analysis. However, Wegner et al. (2015) analyzed size distributions of dust particles in the EDML ice core using a laser particle detector, and found that the mean particle diameter varies between 2–



3 µm. Particles of this size are indeed visible under the microscope and thus our results should be comparable with CFA data. Individual µ-inclusions were selected for Raman measurements. The obtained spectra confirmed that the optically detected "black dots" are in fact real second phase inclusions, mainly salt and dust particles. This supports previous studies by Ohno et al. (2005, 2006) and Sakurai et al. (2011) who used Raman microscopy to analyze µ-particles in ice from Dome Fuji. Detailed

quantitative studies on composition and size of µ-inclusions are ongoing to estimate the actual proportion of substances present as second phase inclusions or dissolved within the ice lattice respectively.

No signal other than ice spectrum could be detected when focusing into grain interiors, grain boundaries or triple junctions. In all four samples presented in our study, impurity spectra could be detected only when focusing onto visible µ-inclusions. Thus we cannot confirm observations by Fukazawa et al. (1998) and Barletta et al. (2012), who measured acidic environments

in triple junctions and grain boundaries via Raman spectroscopy, nor EDX analyses by e.g., Cullen and Baker (2001) and Iliescu and Baker (2008) who also found trace elements in grain boundaries. The AWI cryo-Raman should be considered one of the most powerful Raman systems applied to ice. Still, its spatial resolution and sensitivity are limited by the applied optics and by the physics of Raman scattering. Thus, we cannot rule out segregation of trace elements to grain boundaries if their concentrations would be very low.

High resolution CFA dust-concentrations measured by Wegner et al. (2015) were taken as reference for comparison with our concentrations of µ-inclusions (Tab.1). The ratios dust vs. µ-inclusions differ significantly between the two ice cores. The deep EDML ice in solid state contains 2-3 times more µ-inclusions per volume unit than dust in the melt water (CFA, Wegner et al., 2015). In contrast, the NEEM samples contain comparable amounts of µ-inclusions and dust. However, the NEEM CFA data still need to be flux-calibrated (Wegner, personal communication). Additionally, the presence of air bubbles in the shallower

NEEM samples may cause an underestimation of µ-inclusion concentration. As air inclusions appear dark in the impurity maps (Fig.5), they may cover a substantial part of µ-inclusions in the image. Another possible explanation is to assume that the NEEM samples contain predominantly insoluble dust particles which are detected in the CFA, whereas the EDML µ-inclusions consist mainly of water-soluble substances. The preliminary analysis of the collected Raman spectra points in this direction. While the two Antarctic samples feature primarily sulphate salts (in agreement with Ohno et al., 2006), we mainly found

terrestrial minerals and black carbon in the Greenland samples.

## 4.2 Zener pinning

The attractive force between a grain boundary and second phase inclusions results from the reduction of grain boundary surface energy. Assuming a random distribution of spherical µ-inclusions of radius $r$ (Humphreys and Hatherly, 2004), the maximal pinning pressure on a grain boundary can be derived:

$$P_Z = 2\pi r^2 \gamma N_V \tag{1}$$

where $\gamma$ is the grain boundary surface energy and $N_V$ number of µ-inclusions per volume unit. If the driving force for grain boundary migration $P_{GBM} < P_Z$, then the grain boundary stays in contact with the pinning particles and its migration rate



adapts to the mobility of the particles (slow mode pinning). On the contrary, if $P_{GBM} > P_Z$, the interaction time is short and grain boundary proceeds its motion leaving the particles behind (fast mode pinning).

One of the objectives of our study was to catch µ-inclusions "in flagrante" - i.e. in the very act of pinning a grain boundary. However, this aim resulted to be cumbersome since grain boundaries are usually invisible inside the sample volume. In general,

we can only estimate positions of grain boundaries from the surface images (microstructure maps). Sometimes, if the curvature and convexity are favorable with respect to the image plane, 3D-shapes of grain boundaries are indeed visible within the sample volume. In such cases, we could observe clathrate hydrates sticking to grain boundary interfaces, deforming their shapes due to the pinning force. However, no µ-inclusions were observed to produce such kind of effects. We studied the distribution of µ-inclusions in a plane of focus over the whole sample area. The density of µ-inclusions is inhomogeneous, however exhibits

no correlation with grain boundaries. The typically clustered distribution could in principle be interpreted as caused by sudden release of µ-inclusions from accelerating grain boundary. However, the clusters which are within themselves unordered also are of no significant shape e.g. rows, planes or ellipsoids. We would expect some kind of alignment in rows or planes or at least some sort of graduation (size, type) of the clusters if they would result from release by a moving grain boundary.

The observations listed above lead us to the conclusion, that only fast-mode pinning can take place in all four analyzed

samples:

$$P_{GBM} > P_Z \tag{2}$$

This is in agreement with Alley et al. (1986b); Paterson (1991) and others who suggest that particle concentrations in ice are in general too low to induce slow mode pinning. On the contrary, we cannot confirm the assumption by Durand et al. (2006), that µ-inclusions would accumulate in high concentrations at grain boundaries. Our results also confirm general observations by

Faria et al. (2010) who examined several optical microscopy data sets along the EDML ice core with respect to the integrity of the climatic record. They did not observe redistribution of µ-inclusions due to pinning or dragging down to 2300 m. However, below this depth, and particularly in the deepest 200 meters, the authors found accumulations of "black dots" along grain boundaries and concluded that pinning and dragging is relevant only in the deepest part of the EDML ice core. This finding mismatched our observations in EDML-2371.4 and EDML-2371.9 where we only found secondary bubbles accumulated

along grain boundaries. Since microstructure maps described by Faria et al. (2010) were made only few days or hours after drilling the ice core, we decided to record new microstructure maps of the same spots now after ca. 10 years of storage. The comparison indicates that significant changes occur during the relaxation of the material (see example in Fig.6). The image of the freshly drilled ice shows a group of "black dots" accumulated at a grain boundary. Image of a re-measurement of the same sample shows that with time these "black dots" have grown and filled with gas and now they are forming typical relaxation air

bubbles. There are two possible explanations: 1. Grain boundaries in deep EDML do collect µ-inclusions by dragging but after retrieving the core they serve as seeds for the growing relaxation bubbles and get surrounded with gas. It remains unclear why µ-inclusions in the interior of grains do not evolve into bubbles. 2. Another option is that the "black dots" at grain boundaries observed by Faria et al. (2010) in fact are small micro-bubbles forming due the abrupt drop of pressure after logging the ice core. Weikusat et al. (2012) analyzed compositions of secondary microbubbles using Raman spectroscopy. The authors also





remark that secondary bubbles tend to form at locations of "black dots". However, to date there are no data available concerning the composition of these original "black dots" as Raman spectroscopy is currently not available on-site, viz. right after drilling.

The highest local µ-inclusion density in our sample material was found in the double horizon of sample EDML-2371.9 (Fig.2,3) and could be estimated as $N_V = 37206$ cm$^{-3}$. When inserting this value in Eq. (1), assuming the mean particle radius $r = 1.5$ µm (in agreement with Wegner et al., 2015) and using $\gamma = 65$ mJ m$^{-2}$ for high angle boundaries (Hobbs, 1974), we obtain the maximal pinning pressure $P_Z = 0.034$ N m$^{-2}$. The driving force for grain boundary migration can be written as:

$$P_{GBM} = \Delta H - \frac{2\gamma}{R} \qquad (3)$$

where $\Delta H$ is the gradient in stored strain energy density across the grain boundary (i.e. strain induced driving pressure) and the second term represents the curvature-driven pressure with the radius $R$ of grain boundary local curvature. Assuming no difference in stored strain energy ($\Delta H = 0$), the boundary migration will be driven only by its curvature. Inserting $P_Z$ and Eq. (3) into Eq. (2) we obtain $R < 3.8$ m the radius of local curvature necessary to unpin the grain boundary from its surrounding µ-inclusions. Most grain boundaries along the NEEM and EDML ice cores indeed fullfill this condition (e.g., Binder et al., 2013).

In a second marginal case, let us consider a planar grain boundary ($R \rightarrow \infty$) whose migration is only driven by the difference in stored strain energy as function of the local dislocation density $\Delta\rho_{dis}$. Following Humphreys and Hatherly (2004) and Llorens Verde et al. (2016), $\Delta H$ can be approximated as:

$$\Delta H = \Delta\rho_{dis} \frac{1}{2} G b^2 \qquad (4)$$

where $G b^2 / 2$ is the energy per unit length of a single dislocation line consisting of the shear modulus $G$ and the magnitude of the Burgers vector $b$. Considering only the basal slip system the mean dislocation energy can be estimated as $G b^2 / 2 = 3.6 \cdot 10^{-10}$ J m$^{-1}$. Inserting this in Eq. (4) and combining Eq. (4), (3) and (2) we obtain an estimation for the minimal difference in dislocation density required for unpinning: $\Delta\rho_{dis} > 10^8$ m$^{-2}$. This value is 2–3 orders of magnitude smaller than absolute dislocation densities modeled by Montagnat and Duval (2000) and Montagnat et al. (2003), and also smaller than a minimum dislocation density access calculated from grain boundary curvatures by Hamann et al. (2007, Fig. 11b).

In the above considerations we made a variety of assumptions and the derived values are only for orientation. Furthermore, in the case of a real grain boundary $\Delta H$ acts against the curvature-driven force as described in (3) and thus the final motion and shape are determined by the ratio of these forces. However, the exercise demonstrated that applying Zener's theory to our particular µ-inclusion concentration the pinning effect is comparatively small and will hardly affect grain boundary migration. This is indeed in agreement with what we observe.

## 4.3 Grain size controlling mechanisms

It is difficult to make general conclusions based on the analysis of four discrete samples. Our study indicates that pinning on µ-inclusion occurs in fast mode in large part of the ice sheets, as already stated by Alley et al. (1986b), and will not significantly



affect grain boundary migration. At the same time, with our Raman microscope we find no other form of e.g., dissolved impurities segregated to grain boundaries. However, negative correlations between average grain size and impurity content were found in virtually all ice cores (e.g., Gow and Williamson, 1976; Gow et al., 1997; Lipenkov et al., 1989; Azuma et al., 1999, 2000; Thorsteinsson et al., 1995; Durand et al., 2009). High-impurity ice exhibits generally smaller grains than low-

impurity ice at the same depth. This negative correlation can be found at all scales: in seasonal variabilities (cloudy bands, cm), during rapid climatic fluctuations such as Dansgaard-Oeschger events (tens of meters), or comparing glacial and interglacial periods (hundreds of meters).

    Grain size has an impact on a variety of ice physical properties (Goldsby, 2006; Cuffey, 2006), and vice versa it is controlled by thermodynamic conditions and processes within the ice sheet. Without deformation and under purely static conditions, mean

grain area increases linearly with time. This normal grain growth (NGG) is driven by the reduction of grain boundary surface energy (Alley et al., 1986a) due to the optimization of volume versus interfaces. This model is especially relevant for very small grain sizes well below the equilibrium or steady state grain size (see e.g., Jacka and Jun, 1994; Faria et al., 2014b), such as smallest grain sizes in the uppermost part of the ice sheet (Gow, 1969), but may also apply to some areas where topographic depressions in the bedrock below ice sheets inhibit deformation and NGG can produce extraordinary large grain sizes (stagnant

ice, Budd and Jacka, 1989). If deformation introduces additional energy into the system, dynamic recrystallization processes are activated, driven by the energy reduction. Rotation recrystallization (RRX) splits grains into subgrains and thus has a grain-size-reducing effect (Durand et al., 2008; Alley et al., 1995; Wang et al., 2003). During RRX recovery orders dislocations of a deformed grain into subgrain boundaries, which are lower energy states of dislocation assemblages (Hirth and Lothe, 1982; Hondoh, 2010). Subgrain boundaries can develop into grain boundaries by further rotation. On the contrary, strain-

induced boundary migration (SIBM) is performed by the propagation of grain boundaries instead of dislocation motion. The effectiveness of SIBM increases with the heterogenous distribution of dislocations (Weikusat et al., 2009) in polycrystalline ice, which is a direct consequence of the high mechanical anisotropy of the ice crystal. SIBM leads to huge grain sizes in the deep ice (De La Chapelle et al., 1998), however can also lead to grain size reduction, e.g., by nucleation of new grains (SIBM-N Faria et al., 2014b) or by dissection of highly irregular grains. Recent microstructural studies (e.g., Durand et al.,

2008; Faria et al., 2014a) show that all recrystallization mechanisms (NGG, RRX and SIBM) concur all over the depth range of an ice sheet, rather than being dominantly active in separate depth zones. The grain size is then a product of the interplay between these processes (dynamic grain growth, Faria et al., 2014b).

    It is widely accepted that the grain size is modulated by some impurity effect. The two most manifest candidates for such an effect are the reduction of grain boundary mobility via dissolved impurities (Alley et al., 1986b; Alley and Woods, 1996;

Paterson, 1991) and grain boundary pinning by μ-inclusions (Fisher and Koerner, 1986; Durand et al., 2006). However, experimental evidences for both models are controversial, as demonstrated by our study. Furthermore, both interpretations are based on the assumption that it is solely NGG which suffers under the effect of impurities leading to smaller grain sizes. However, grain size is a product of all recrystallization processes together, as discussed in previous paragraph. Therefore we hypothesize that an indirect impurity effect, for instance enhanced deformation and/or effect on dynamic recrystallization triggered by high

impurity content could be an alternative candidate responsible for the changes in grain size. Impurities could have a significant





influence on strain distribution within grains as well as dislocation mobilities and densities, e.g., via dislocation multiplication. The microstructure effects of increasing dynamic recrystallization versus viscoplastic deformation have recently been tested by a microstructural evolution model (Llorens 2016 a,b). In these model runs the impact is significant on e.g. the grain shape evolution, because deformation tends to flatten grains while recrystallization tends to make them equidimensional. First evidences

of changes in the deformation-recrystallization interplay have been observed by means of grain shape analyses (Weikusat et al., 2016; Binder et al., 2013). However, to proof or disclaim these hypotheses is far beyond the scope of this study.

## 5   Summary

We present high resolution high-resolution large-scale maps (3 µm pix$^{-1}$, $8 \times 2$ cm$^2$) of µ-inclusions within four samples from polar ice cores - two from the EDML (2371 m) and two from the NEEM ice core (740 m). For the first time, in-situ distributions

of a representative number (more than 5000) of µ-inclusions have been studied. A confocal Raman microscope has been used to prove the impurity origin of the inclusions. Discrete µ-inclusions are the only impurity form detected in this study, i.e., we measured no signal attributed to dissolved impurities neither in grain interiors nor in grain boundaries. The comparison with grain boundary network shows no correlation between µ-inclusions and grain boundaries. Therefore we observed no evidence for redistribution of impurities by dragging, nor for slow mode grain boundary pinning as defined by Alley et al. (1986b). The

link between grain size and impurities may not be (only) due to hindered normal grain growth in impurity-rich ice. Deformation and dynamic recrystallization enhanced by impurities possibly also have a grain-size-reducing effect.

*Author contributions.* J.E., I.K. and W.S. performed the measurements, processing and interpretation. M.B., D.J., S.K., C.W., I.W. supported the processing and interpretation of data by providing specialists knowledge as well as the general glaciological framework. I.W. provided the initial concept. Under the lead of J.E. all authors contributed to writing of the manuscript.

*Acknowledgements.* This research was funded by HGF grant VH-NG-802 to J.E. and I.W., SPP 1158 DFG grant WE4711/2 to C.W., as well as DFG grant SPP 1158 BA 3694/2-1 and JSPS fellowship ID PE16746 to M.B. The Microdynamics of Ice (MicroDICE) research network, funded by the European Science Foundation, is acknowledged for funding research visits of J.E. and I.K. (short visit grant). We thank Anna Wegner, Melanie Behrens and Maria Hörhold for discussions on solubility of impurities and CFA related issues. The visual stratigraphy line scan image has been made available at www.pangaea.de. We thank the logistics and drilling team of the Kohnen and NEEM stations. NEEM

is directed and organised by the Center of Ice and Climate at the Niels Bohr Institute and US NSF, Office of Polar Programs. It is supported by funding agencies and institutions in Belgium (FNRS-CFB and FWO), Canada (NRCan/GSC), China (CAS), Denmark (FIST), France (IPEV, CNRS/INSU, CEA and ANR), Germany (AWI), Iceland (RannIs), Japan (NIPR), Korea (KOPRI), the Netherlands (NWO/ALW), Sweden (VR), Switzerland (SNF), UK (NERC) and the USA (US NSF, Office of Polar Programs). This work is a contribution to the European Project for Ice Coring in Antarctica (EPICA), a joint European Science Foundation/ European Commission (EC) scientific programme, funded by

the EC and by national contributions from Belgium, Denmark, France, Germany, Italy, The Netherlands, Norway, Sweden, Switzerland and the UK.



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



| ice core | EDML | | NEEM | |
|---|---|---|---|---|
| sample | 2371.9 | 2371.4 | 1346.2 | 1346.5 |
| depth (m) | 2370.9 | 2370.4 | 739.9 | 740.2 |
| sample size (mm) | 76 × 17 | 80 × 21 | 71 × 21 | 74 × 23 |
| total number of μ-inclusions | 2527 | 1195 | 1145 | 917 |
| ice density (g/ml) | 0.9167 | 0.9167 | 0.9079 | 0.9079 |
| number of μ-inclusions / ml water | 10668 | 4972 | 4197 | 3019 |
| dust particles / ml water (CFA) | 3575 | 2645 | 5450 | 3823 |

**Table 1.** Black dots from optical microscopy versus dust concentration from CFA (Wegner et al., 2015). With the sample size, the focus range of 200 μm and the density of the ice we estimate the concentration of μ-inclusions. The density of the samples has been estimated using images of air bubble density.

| ice core | EDML | | NEEM | |
|---|---|---|---|---|
| sample | 2371.9 | 2371.4 | 1346.2 | 1346.5 |
| depth (m) | 2370.9 | 2370.4 | 739.9 | 740.2 |
| total number of μ-inclusions | 2527 | 1195 | 1145 | 917 |
| μ-inclusions within 300 μm around a GB | 183 | 127 | 278 | 164 |
| percentage | 7% | 11% | 24% | 18% |

**Table 2.** Counted μ-inclusions in the vicinity of grain boundaries (region of 300 μm thickness along grain boundaries) obtained from stacked microstructure maps with grain boundary grooves on the surface and impurity maps focused inside the sample (ca. 300μm below the sample surface).





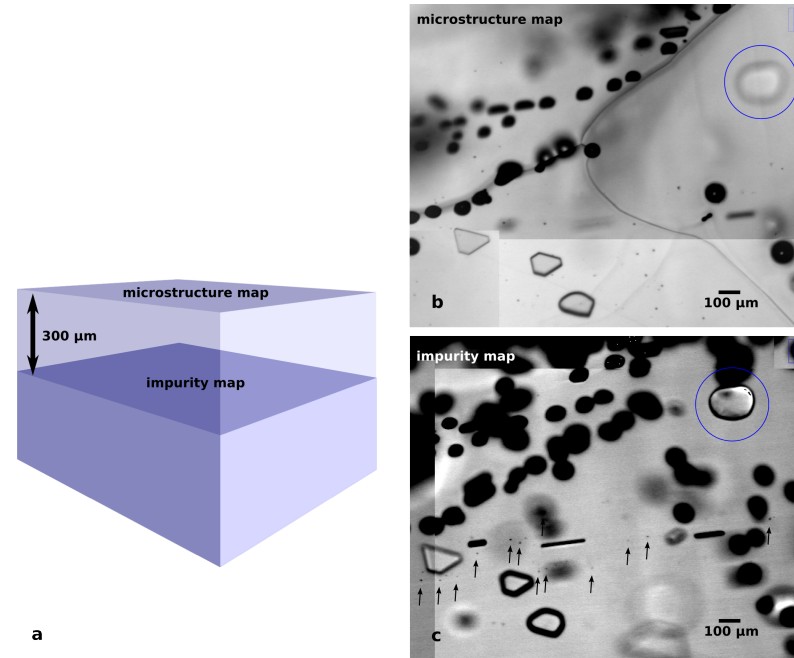

**Figure 1.** (a) Schematic of positioning stacked microstructure maps from surface and inside the sample. The focus distance between the two mapping planes in general depends on the application of this method (e.g. investigation of air inclusions or µ-inclusions). In order to correlate µ-inclusions and grain boundary grooves, for this study a distance of 300µm has been chosen. (b) Surface-focused photomicrograph from EDML-2371.9 (2371 m, drilled in 2006). The image shows three grain boundaries as black thin curves and a triple junction. "Black dots" in this map type are due to surface pollution by frost particles and should not be confused with µ-inclusions in the impurity maps. Below the surface - and slightly out of focus - a clathrate hydrate (marked with blue circle) and three Plate-like inclusions (hexagonal objects) are visible. Roundish black objects of the size of up to 100 µm are secondary gas inclusions formed by relaxation of the material. At the surface they are clearly attached to the left grain boundary. (c) Photomicrograph focused ca. 300µm below the surface. The grain boundary grooves disappeared but objects in the sample volume came into focus - the clathrate hydrate and plate-like inclusions are sharp now. "Black dots" in this image (pinpointed with arrows) are chemical impurities in form of µ-inclusions.




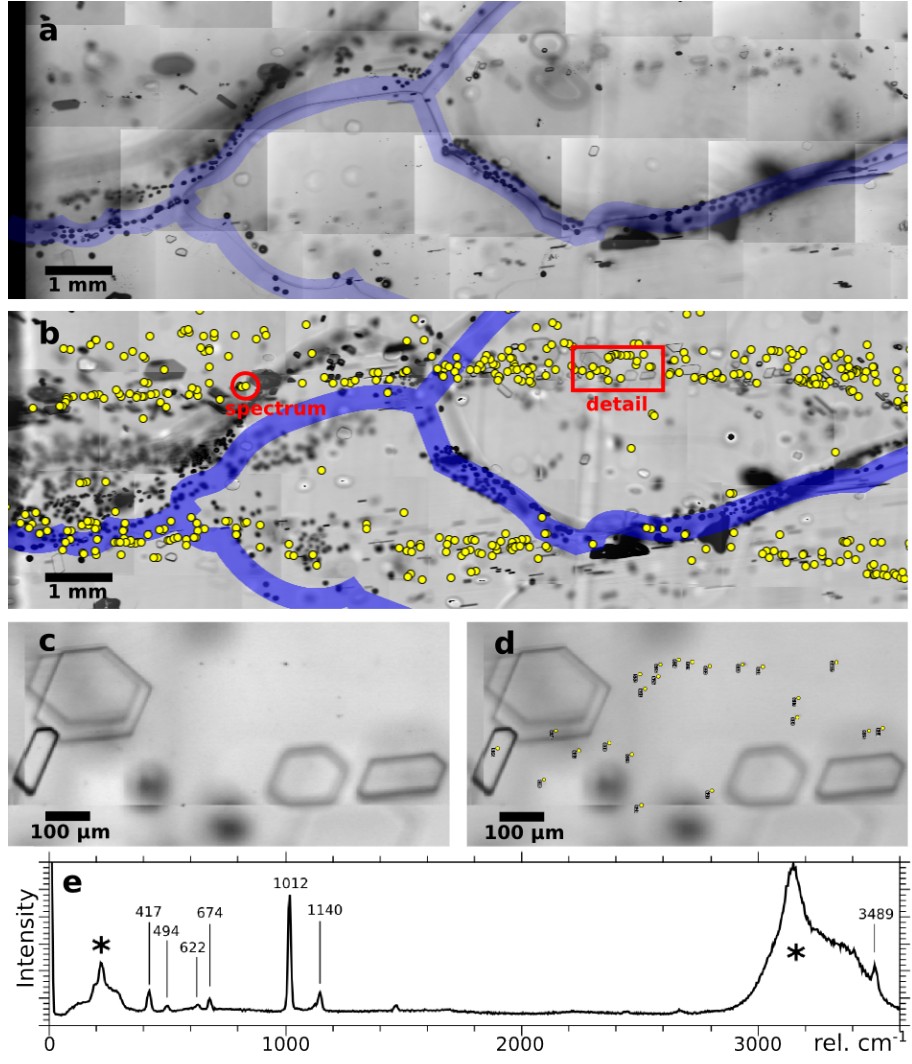

**Figure 2.** Double impurity layer in EDML-2371.9 (horizon 1 in Fig.3). (a) Surface map with grain boundaries visible as thin dark lines highlighted with blue bands. Blurred lines are grain boundaries at the bottom side of the specimen which are out of camera focus. (b) Impurity map with marked μ-inclusions (yellow circles) and grain boundary network from the surface map. Roundish black objects with the size of several tens of μm are secondary gas inclusions (micro-bubbles) formed due to relaxation (Weikusat et al., 2012). While μ-inclusions follow horizontal layering, micro-bubbles trace the 3D shape of the grain boundaries. (c) A high resolution uninterpreted detail of the upper impurity layer - red rectangle in (b). (d) Same detail with interpreted μ-inclusions. (e) Example Raman spectrum of one μ-inclusion from the upper layer (red circle). Parts of the signal marked with asterisk correspond to the ice spectrum. The positions of the proper peaks are quoted. The inclusion is a gypsum particle ($CaSO_4 \cdot 2H_2O$).




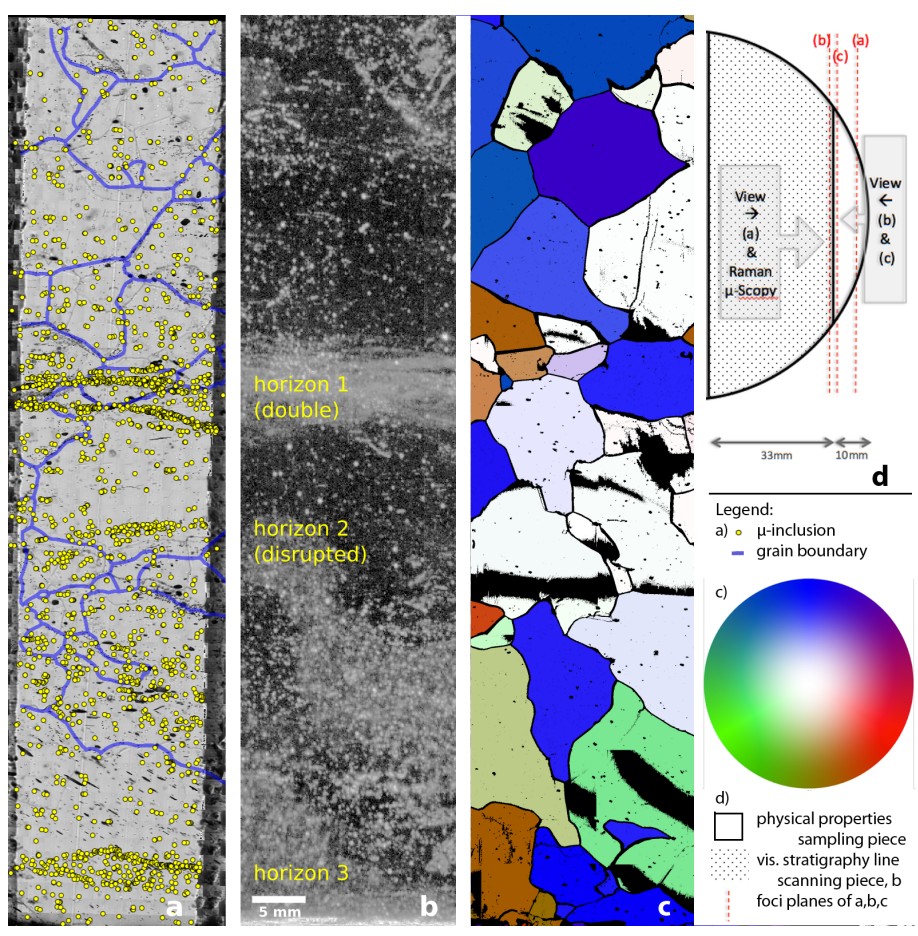

**Figure 3.** EDML-2371.9: (a) Impurity map of the whole sample (17 mm × 76 mm) with 2527 μ-inclusion (yellow circles) and grain boundaries from the surface map (blue bands). Blurred lines are grain boundaries at the bottom side of the specimen. Horizontal layers (1–3) of μ-inclusion are clearly visible. (b) A detail from the linescanner image of the same part of the ice core. Horizon 1 is visible as a cloudy band. (c) C-axis orientations of individual grains projected into a horizontal plane. Vertical c-axes appear white, horizontal c-axes appear in full colors depending on their azimuth (see color code in the legend). (d) Cutting plan of the sample preparation.



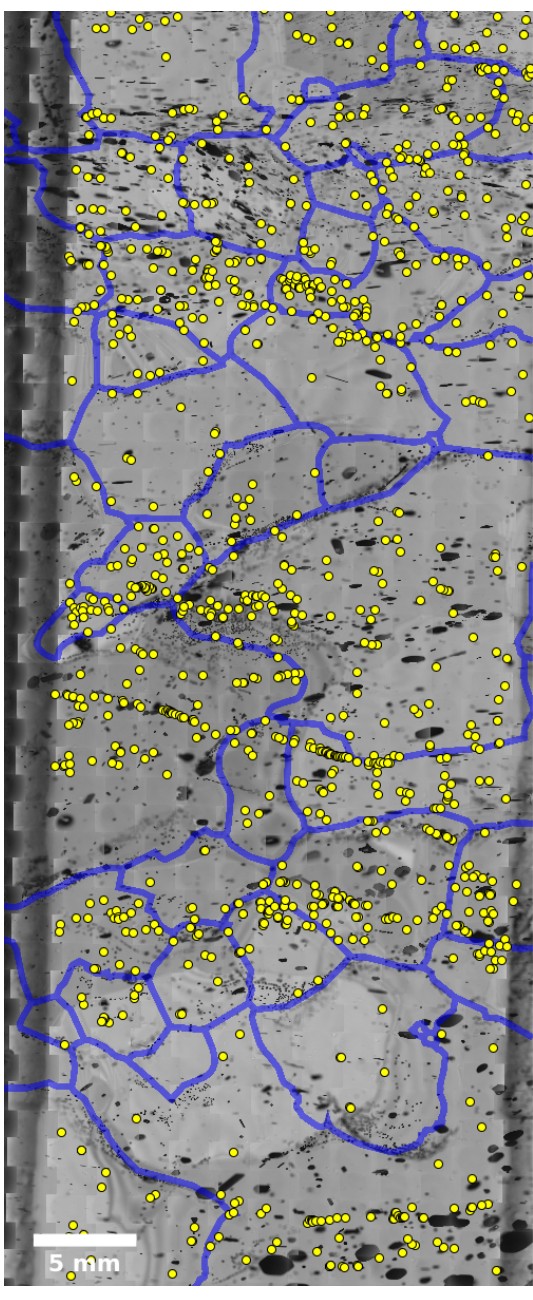

**Figure 4.** Microstructure and impurity map of the sample EDML-2371.4. Layering of μ-inclusions (yellow) is less pronounced than in EDML-2371.9. Secondary gas inclusions (small, black and roundish) tend to follow the shapes of grain boundaries. In contrary, the μ-inclusions seem not to accumulate at grain boundaries.





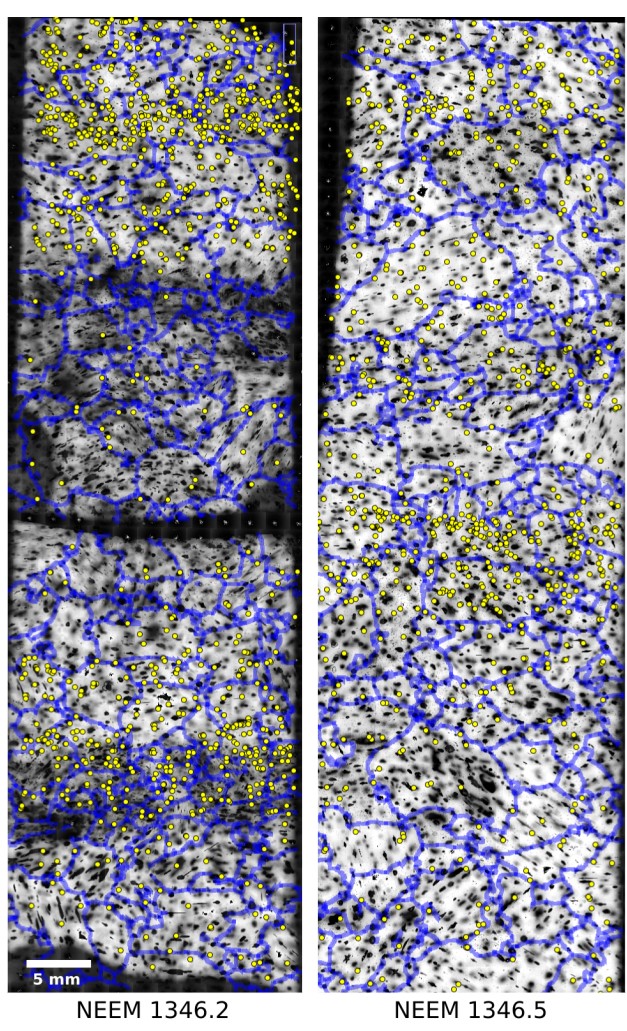

NEEM 1346.2          NEEM 1346.5

**Figure 5.** Microstructure and impurity maps of the two NEEM samples (740 m). Horizontal layering of µ-inclusions is present in both maps. The samples contain a higher density of grain boundaries, the average grain radius is 1.5 mm. NEEM-1346.2 is cracked in the central part. In contrast to the EDML samples, the gas inclusions (black) are homogeneously distributed, since they are primary air bubbles, albeit deformed due to relaxation.



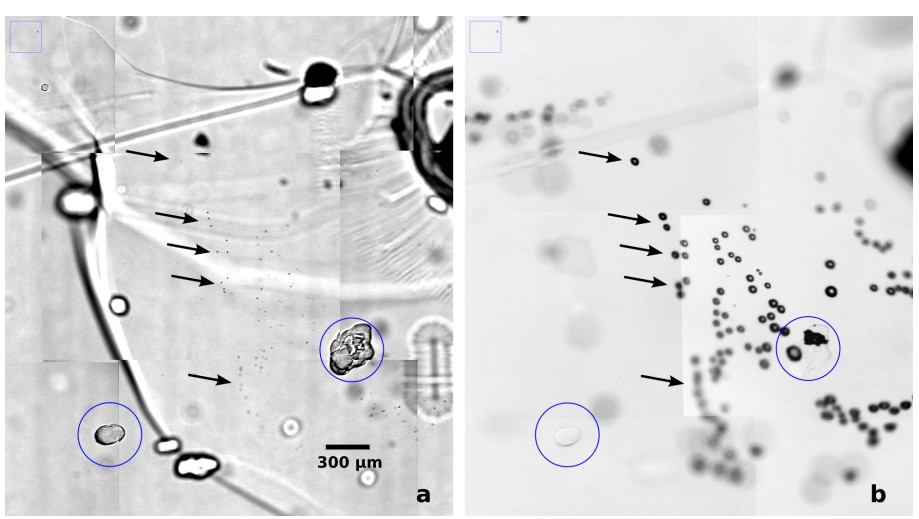

**Figure 6.** Two images of the same spot in EDML 2376.0. Two clathrate hydrates are highlighted (blue circles) in both images for comparison.
a) Photomicrograph taken at the site immediately after drilling the ice core, 2006. An accumulation of "black dots" within the grain boundary plane can be recognized. These "black dots" are slightly larger than the μ-inclusions counted in our impurity maps. b) The same spot after ca. 10 years of relaxation. Almost all "black dots" at the grain boundary evolved into secondary gas inclusions.