# Peer review of "Location and distribution of micro-inclusions in the EDML and NEEM ice cores using optical microscopy and in-situ Raman spectroscopy"

_The Cryosphere, 2016_

## Referee Comment (RC1) · A. Svensson (Referee) · 11 Jan 2017

General comments:

The manuscript describes the distribution and properties of individual micro-impurities in four ice core samples from the Antarctic EDML and the Greenland NEEM ice cores. The dataset is compared to various other impurity datasets and the implications of findings are discussed.

The study is very solid, carefully carried out, and well presented. The issue of impurities in natural ice is complex and has importance for a variety of ice and climate related fields as mentioned in the introduction. The presented work pushes the discussion one

important step forward and it fits well within the scope of TC.

Whereas I appreciate the time-consuming experimental work presented in the manuscript, the weakness of the study clearly is the very limited diversity of analyzed samples. Only samples of very 'clean' ice-sheet ice have been analyzed and so we are still not able to answer important questions about the location of ice core impurities within the ice structure, possible pinning, and ice flow in more 'dirty' glacial ice. I suspect, however, this will be the topic of forthcoming papers by the authors now the technique has been firmly established.

Specific comments:

I would mention the climatic periods the samples are obtained from in the abstract. NGRIP Holocene and EDML MIS 5.5

Please define what is meant by 'second phase inclusions' that are mentioned several times.

Would you have a reference for the statement on p. 3 l. 14-15 that CPO varies with impurity concentrations of the ice?

You refer to a study by Della Lunga et al., 2014, where impurities are stated to be found in grain boundaries in 'clean' ice. Could you have a more quantitative estimate of the 'cleanness' of those samples as compared to the ice investigated in the present study? In other words, are the samples of the two studies directly comparable or could it potentially be that the conclusions of both studies are correct?

As support to the conclusion of the manuscript, that impurities stay within the ice at the position where they are deposited rather than being dragged around by migrating crystal boundaries, I would like to mention a paper of my own (Svensson et al., 2011), where we identify clear annual layering in impurities within large ice crystals of NGRIP Eemian ice. In this clean ice the annual banding, and thus the micro-inclusions I suspect, is sometimes visible to the naked eye.

In my opinion, the comparison of the micro-inclusions to that of CFA records could be somewhat improved. First, the micro-inclusions include both insoluble dust and chemical particles. Therefore, it would make sense to compare the distribution of micro-inclusions not only to insoluble dust, but also to chemical concentrations such as Ca, if they are available. Second, if I got it right, you need the micro-inclusions to be larger than 2-3 micros in order to detect them? The Abacus CFA dust analyzer, however, will typically detect particles larger than 1 micron. With the insoluble dust size distribution centered around a few microns you are likely to miss out a large fraction of the dust particles in your observations. Those two issues taken together, it is not surprising that the numbers in Table 1 obtained by your method and by the Abacus do not agree so well. A way of making a more detailed comparison of your counts to that of the CFA records would be to plot the CFA profiles on a depth scale along with the samples in Figures 3+4+5 or alternatively in a separate figure. On p. 6 l. 28 you state that you have made this comparison, so why not show it? One question is of course if the CFA profiles have sufficiently high depth resolution to resolve the details of the banding in your samples.

It would also be of interest if you could provide an estimate of the annual layer thickness of the investigated ice sections. Is the stratigraphy of the micro-inclusions, as evidenced in figures 3, 4, and 5, related to the seasonal variations of impurity deposition?

P. 6. l. 1, please define z-position.

Figure 3: Why is there apparently such poor correspondence between the derived crystal boundaries in a) and c) ?

Reference:

Svensson, A., Bigler, M., Kettner, E., Dahl-Jensen, D., Johnsen, S., Kipfstuhl, S., Nielsen, M., and Steffensen, J. P.: Annual layering in the NGRIP ice core during the Eemian, Clim. Past., 7, 1427-1437, 2011.

---

## Referee Comment (RC2) · J. L. Urai (Referee) · 14 Feb 2017

[referee-annotated manuscript omitted]

---

## Author Comment (AC1) · 13 Mar 2017

Many thanks to our referees for their constructive feedback. Our answers to their comments as well as the changes made in the manuscript are summarized in a pdf file included in the supplement. Also a marked-up manuscript version is attached.

Please also note the supplement to this comment:
http://www.the-cryosphere-discuss.net/tc-2016-247/tc-2016-247-AC1-supplement.zip

---

## Author Response (AR1)

**Author's response to interactive comments on**

**"Location and distribution of micro-inclusions in the EDML and NEEM ice cores using optical microscopy and in-situ Raman spectroscopy" by J. Eichler et al.**

**Referee 1: A. Svensson**

**General comments:**
The manuscript describes the distribution and properties of individual micro-impurities in four ice core samples from the Antarctic EDML and the Greenland NEEM ice cores. The dataset is compared to various other impurity datasets and the implications of findings are discussed.

The study is very solid, carefully carried out, and well presented. The issue of impurities in natural ice is complex and has importance for a variety of ice and climate related fields as mentioned in the introduction. The presented work pushes the discussion one important step forward and it fits well within the scope of TC.

Whereas I appreciate the time-consuming experimental work presented in the manuscript, the weakness of the study clearly is the very limited diversity of analyzed samples. Only samples of very 'clean' ice-sheet ice have been analyzed and so we are still not able to answer important questions about the location of ice core impurities within the ice structure, possible pinning, and ice flow in more 'dirty' glacial ice. I suspect, however, this will be the topic of forthcoming papers by the authors now the
technique has been firmly established.

Thank you for reviewing the manuscript and for your constructive and motivating feedback. I agree with most of your objections and considered them in the corrected manuscript version. Indeed, all four analyzed samples are from the 'clean' interglacial parts of the ice cores, which makes general statements difficult. We actually had a look into 'dirty' glacial ice, however a large-area scan comparable to the presented data was not performed yet. Therefore the manuscript focuses upon interglacial ice.

**Specific comments:**
I would mention the climatic periods the samples are obtained from in the abstract. NGRIP Holocene and EDML MIS 5.5

We added the climatic periods to the abstract.
Change: p. 1 l. 7-8.

Please define what is meant by 'second phase inclusions' that are mentioned several times.
Second phase inclusions or second phase particles refer to inclusions of different material or phase, incoherent with the host matrix. The term appears in the literature (e.g., Ashby 1969, Alley 1986, Humphreys and Hatherly 2004). At various positions in the manuscript it was used as a synonym for micro-inclusions, which was probably confusing. We modified the text and checked for consistency.
Change: p. 1 l. 3, p. 3 l. 24, p. 4 l. 23-24, p. 5 l. 28, p. 8 l. 17.

Would you have a reference for the statement on p. 3 l. 14-15 that CPO varies with impurity concentrations of the ice?

The variation of CPO with impurity content occurs at transitions between glacial and interglacial periods (e.g., Gow and Williamson, 1976; Paterson, 1991; Faria et al., 2014a ). However, on a small scale (cloudy bands), CPO correlations with impurities are rare.
Change: p. 3 l. 20-23

You refer to a study by Della Lunga et al., 2014, where impurities are stated to be found in grain boundaries in 'clean' ice. Could you have a more quantitative estimate of the 'cleanness' of those samples as compared to the ice investigated in the present study? In other words, are the samples of the two studies directly comparable or could it potentially be that the conclusions of both studies are correct?

Indeed the samples are not directly comparable. Both rather show case studies of different ice (glacial versus interglacial) with different methods (Raman versus LA-ICPMS). We changed the text accordingly to explain this.
Change: p. 3 l. 5-8, p. 11 l. 4-10.

As support to the conclusion of the manuscript, that impurities stay within the ice at the position where they are deposited rather than being dragged around by migrating crystal boundaries, I would like to mention a paper of my own (Svensson et al., 2011), where we identify clear annual layering in impurities within large ice crystals of NGRIP Eemian ice. In this clean ice the annual banding, and thus the micro-inclusions I suspect, is sometimes visible to the naked eye.

Thank you very much for bringing this paper to our attention. Indeed it supports our conclusions. We added the reference.
Change: p. 10 l. 18-21.

In my opinion, the comparison of the micro-inclusions to that of CFA records could be somewhat improved. First, the micro-inclusions include both insoluble dust and chemical particles. Therefore, it would make sense to compare the distribution of micro-inclusions not only to insoluble dust, but also to chemical concentrations such as Ca, if they are available. Second, if I got it right, you need the micro-inclusions to be larger than 2-3 micros in order to detect them? The Abacus CFA dust analyzer, however, will typically detect particles larger than 1 micron. With the insoluble dust size distribution
centered around a few microns you are likely to miss out a large fraction of the dust particles in your observations. Those two issues taken together, it is not surprising that the numbers in Table 1 obtained by your method and by the Abacus do not agree so well. A way of making a more detailed comparison of your counts to that of the CFA records would be to plot the CFA profiles on a depth scale along with the samples in Figures 3+4+5 or alternatively in a separate figure. On p. 6 l. 28 you state that you have made this comparison, so why not show it? One question is of course if the CFA profiles have sufficiently high depth resolution to resolve the details of the banding in your samples. It would also be of interest if you could provide an estimate of the annual layer thickness of the investigated ice sections. Is the stratigraphy of the micro-inclusions, as evidenced in figures 3, 4, and 5, related to the seasonal variations of impurity deposition?

We improved the comparison with CFA. The formulation regarding resolution limit of the microscope was probably misleading. Both the light microscope and the Raman microscope are in principle able to resolve particles of 1 micron in diameter, i.e. the same size as the

Abacus dust analyzer. The impurity maps as presented in Fig.3,4,5 should therefore contain (almost) the whole size spectra. Thus we believe that the concentration differences presented in Table 1 are mainly caused by the water-soluble fraction of micro-inclusions, rather than by systematically missing out a whole fraction of small-sized particles.
We added an estimate of the annual layer thickness and discuss the relation between micro-inclusion layers and seasonal variations.
We added a new figure (Fig.4) with the high resolution CFA profile of the whole EDML bag 2371 and a paragraph discussing correlations between dust, Ca and conductivity signal.
Change: p. 8 l. 9-16, p. 9 l. 8-24, p. 24 (Fig.4).

P. 6. l. 1, please define z-position.
We changed the term z-position to z-coordinate and defined it as the distance from the sample surface.
Change: p. 6 l. 11-12.

Figure 3: Why is there apparently such poor correspondence between the derived crystal boundaries in a) and c) ?
The poor correspondence between grain boundaries in Fig.3 a) and c) is caused by the relatively large distance between the two cuts of the sample. This should be illustrated with the cutting plan d). We added a short explanation in the figure caption.
Change: p. 23 caption Fig. 3.

**Referee 2: J. Urai**

Thank you for your review of our manuscript. We implemented all your suggestions regarding orthography and better suited expressions, in particular:
p. 1, l. 3, 11, 18
p. 2, l. 11, 25
p. 6, l. 18
p. 7, l. 20
p. 9, l. 27
p. 11, l. 33
p. 12, l. 26-27
p. 13, l. 14.

Perhaps it is worth considering here the effect of impurities on grain boundary mobility.
Drury, M.R. and Urai, J.L. (1990).
Urai, J.L., Means, W.D. and Lister, G.S. (1986).
Urai, J.L. and Jessell, M. (2001).
Thank you for the comment, we are mentioning the reduction of grain boundary mobility via dissolved impurities as one of the two most manifest mechanisms. In order to point out and increase the visibility in the corrected version we numbered the two arguments: 1. GB mobility via dissolved impurities, 2. Zener pinning by micro-particles. Since our methods do not permit us to access dissolved impurity traces in grain boundaries (probably due to their very low concentrations) we do not further deepen the discussion of their effect.
Also thank you for the references, we included the first two of them due to their high relevance for our topic.

Change: p. 13, l. 2-6.

Structures and fabrics in glacial ice: A review
Peter J. Hudleston - Journal of Structural Geology 81 (2015) 1e27
is this relevant and worth citing?
Thank you for drawing our attention to this very nice publication. However, we believe that the focus of the paper lies too far from our topic.

**Other changes made**
Co-author's last name corrected: Bayer-Giraldi.
Added Legrand1997 to reference in p. 1 l. 15.
Use spelling "sulfate".
Update reference status Weikusat et al 2016: Physical analysis of an Antarctic ice core...

[revised manuscript text omitted]